# Morphology and Distribution of the Antennal Sensilla of Two Species, *Megalurothrips usitatus* and *Thrips palmi* (Thysanoptera: Thripidae)

**DOI:** 10.3390/insects10080251

**Published:** 2019-08-15

**Authors:** Xiao-Shuang Wang, Ali Shaukat, Yun Han, Bo Yang, Liang-De Tang, Jian-Hui Wu

**Affiliations:** 1Key Laboratory of Bio-Pesticide Innovation and Application, Engineering Technology Research Center of Pest Biocontrol, College of Agriculture, South China Agricultural University, Guangzhou 510642, China; 2Key Laboratory of Integrated Pest Management on Tropical Crops, Ministry of Agriculture, Environment and Plant Protection Institute, Chinese Academy of Tropical Agricultural Sciences, Haikou 571101, China

**Keywords:** *Megalurothrips usitatus* Bagnall, *Thrips palmi* Karny, scanning electron microscopy, antennal sensilla

## Abstract

The morphology and distribution of the antennal sensilla of *Megalurothrips usitatus* Bagnall and *Thrips palmi* Karny were examined using scanning electron microscopy (SEM). These are serious pests of various economically important crops, and their antennae are important in chemical communication. The antennae of both species consist of a scape, pedicel, and flagellum, but the flagellum of *M. usitatus* is made up of six sub-segments, whereas that of *T. palmi* consists of five sub-segments. Seven morphological sensilla types, including Böhm bristle (BB), sensilla campaniformia (Sca), three types of sensilla basiconica (Sb1, Sb2 and Sb3), two types of sensilla chaetica (Sch1 and Sch2), sensilla styloconica (Sst), sensilla trichodea (St), and sensilla cavity (Scav), were recorded in both species. The scape and pedicels exhibited Sch1, BB and Sca. The flagellum exhibited two types of Sch, three types of Sb, St, Sst and Scav. Based on these results, the putative function of the sensilla of *M. usitatus* and *T. palmi* are also discussed.

## 1. Introduction

A few species belonging to the family Thripidae are economically important pests of various crops. The bean flower thrips, *Megalurothrips usitatus* Bagnall, is a destructive insect pest of leguminous crops (Fabales: Fabaceae) [1]. The melon thrips, *Thrips palmi* Karny, is an important insect pest of crops belonging to the families Cucurbitaceae and Solanaceae. *T. palmi* is also known for transmitting three tospoviruses, namely the watermelon silver mottle virus (WMSMoV), groundnut bud necrosis virus (GBNV), and melon spotted wilt virus [2]. These two species of thrips can pierce plant tissues and actively suck up their contents during the feeding process [3]. While direct damage, such as necrosis and the premature dropping of buds and flowers, due to thrips feeding and ovipositing, can occur, it is the transmission of viruses that frequently causes the most severe damage to crops [3]. The method by which Thysanoptera species feed allows the thrips to respond to chemical cues, such as plant volatiles and pheromones, during host location, feeding, oviposition, and mating [4].

Sensilla, often referred to as “sensoria” by many thysanopterists, are cuticular sensory organs consisting of a complex of bipolar neurons and enveloping cells [5]. They are important structures for gathering information from their abiotic and biotic environment [6], and they play an important role in various behavioral activities, including host location, host recognition, feeding, mating, and oviposition [7,8,9,10]. They can be found on different body parts, but mainly on antennae and mouthparts [5]. In view of their concentrated area of sensilla, especially chemosensilla, antennae play a crucial role in the behavior of insects and their response to both chemical and physical stimuli [11]. Insects mainly rely on antennal sensilla to perceive volatile organic compounds (VOCs). Different types of olfaction sensilla are distributed on the antennae of insects, so understanding the antennal sensilla is crucial to studies of the host plant selection and feeding behavior.

Thysanoptera are tiny insects, and the use of electron microscope (EM) is an obligatory tool when pursuing a high-quality morphological investigation [5]. In recent years, with the development of EM technology, scanning electron microscopy (SEM) has been extensively used, especially to improve the morphological description of thrips. The antennal sensilla of various species of thrips have been characterized by using EM techniques, including *Bagnalliella yuccae*, *Frankliniella tritici*, *Kladothrips intermedius*, *Koptothrips dyskritus* and other species of thrips [5,12]. Nevertheless, surveys on the antennal sensilla of Thysanoptera are still sparse compared to other insect orders.

It is nevertheless interesting to notice that, while many studies have focused on the biological characteristics and control strategies of *M. usitatus* and *T. palmi* [13,14,15,16,17,18,19], only a few have focused on aspects of the peripheral chemosensory system, such as sensilla. With this study, we managed to offer the first complete ultrastructural survey of antennal sensilla in *M. usitatus* and *T. palmi*, using SEM, and we offer a basis for further investigations, e.g., electrophysiological studies.

## 2. Materials and Methods

### 2.1. Insects

*M. usitatus* were collected from the cowpea crop (*Vigna unguiculata* (Linn.) Walp) grown in Chengmai City, Hainan Province, China (N 19°44′25.47″, E 110°0′2.31″). *T. palmi* were collected from the eggplant crop (*Solanum melongena* L.) grown in Sanya City, Hainan Province, China (N 18°15′15.04″, E 109°30′28.80″). In the laboratory, *M. usitatus* and *T. palmi* colonies were reared on fresh cowpea pods and eggplant leaves, respectively, at 26 ± 1 °C, 70 ± 5% RH, and 14 L: 10 D in a climate control chamber.

### 2.2. Scanning Electron Microscopy (SEM)

The living specimens were decapitated under a continuous variable times stereomicroscope (Stemi 508, Zeiss, Germany), and the heads were fixed in 2.5% glutaraldehyde for 6 h at 4 °C. After rinsing them thrice for 10 min each time with 0.1 M PBS buffer, they were dehydrated in an ascending series of ethanol (30%, 50%, 70%, 80%, 85%, 90%, and 95%) for 20 min at each concentration. Then, they were completely dehydrated twice in 100% ethanol for 20 min each time. Subsequently, the samples were rinsed twice in 100% acetone. After air drying, the samples were critical-point dried (K850, Quorum, England) and mounted with the dorsal-side, lateral-side, and ventral-side on the surface of aluminous stubs with double-sided sticky tape. The sputter coating with gold/palladium was performed with a sputter coater (SC7620, Quorum, England) and subsequently examined with a scanning electron microscope (ΣIGMA, Carl Zeiss, Germany) at an accelerating voltage of 2–6 kV.

### 2.3. Statistical Analysis

The lengths and diameters of the antennae and antennal sensilla of the samples were measured from SEM images using Adobe Photoshop (version CS). Significant differences between the measured parameters from males and females were determined using independent sample T-tests, performed using SPSS 17.0. For the terminology and classification of antennal sensilla used here, we refer to Schneider [20] and Zacharuk [21].

## 3. Results

### 3.1. General Morphology of Antennae

The analysis of micrographs confirms that the antennae of *M. usitatus* (Figure 1a) and *T. palmi* (Figure 1b) do not differ substantially, at least on a morphological level. The antennae of both species are moniliform in shape and consist of a scape, pedicel, and flagellum. It is noted that the flagellum of *M. usitatus* is made up of six sub-segments, whereas the flagellum of *T. palmi* consists of five sub-segments. The antenna of *M. usitatus* is approximately 320 μm long in females and 253 μm long in males (Table 1), and the antenna of *T. palmi* is approximately 214 μm long in females and 213 μm long in males (Table 1). There was a consistent sexual difference (*p* < 0.05) in *M. usitatus* for all antennal segments and total length, but not so in *T. palmi*.

### 3.2. Types of Antennal Sensilla

Seven different morphological types of sensilla on the antennae of *M. usitatus* and *T. palmi* were Böhm bristle (BB), sensilla campaniformia (Sca), sensilla trichodea (St), three types of sensilla basiconica (Sb1, Sb2 and Sb3), two types of sensilla chaetica (Sch1 and Sch2), sensilla styloconica (Sst), and sensilla cavity (Scav). There was no clear sexual dimorphism in the number and distribution of the antennal sensilla in males and females. The scape and pedicel were sparsely covered with Sch1, BB and Sca. The flagellum exhibited two types of Sch, three types of Sb, St, Sst and Scav. The number and distribution of antennal sensilla are summarized in Table 2. The length and diameter of antennal sensilla are summarized in Table 3 and Table 4, respectively. 

#### 3.2.1. Böhm Bristles (BB)

Böhm bristles were observed on the base of the scape and the joints between the scape and pedicel. Böhm bristles are a thorn-like structure with the base inserted in a circular pocket. Their walls are smooth with no pores or grooves on the surface that tapers to a blunt apex (Figure 2a,b and Figure 3d). Five BBs were found on the antennae of both species (Table 2). The mean length of the BB is 3.04 μm in female *M. usitatus* and 2.59 μm in female *T. palmi* (Table 3). The proximal diameter and middle diameter is 0.81 μm and 0.38 μm respectively in female *M. usitatus* (Table 4).

#### 3.2.2. Sensilla Campaniformia (Sca)

Sensilla campaniformia were only found on the distal part of the pedicel ventral surface of both thrips species (Table 2). The pedicels bore a single Sca. These sensilla reached slightly above the antennal surface. They are elliptical with smooth cuticles and elevate thick circular rims (Figure 2c,d,j and Figure 3a). The circular collars of female *M. usitatus* and female *T. palmi* are 3.86 μm and 3.75 μm in diameter, respectively (Table 3). 

#### 3.2.3. Sensilla Trichodea (St)

Sensilla trichodea were observed on the termini of antennae, arising directly from the antennal surface with no socket and slightly elevated above the cuticle. These sensilla are long and gradually curved distally, with a pointed tip. Their walls are smooth, with no pores or grooves on the surface (Figure 2i). The length of these sensilla is approximately 19.34 μm (Table 3), whereas their proximal diameter is 1.16 μm in the female *M. usitatus* (Table 4).

#### 3.2.4. Sensilla Chaetica (Sch)

Sensilla chaetica were observed on each antennal segment of both species (Table 2). These sensilla are straight or slightly curved, longitudinally grooved, and have a sharp or blunt tip. This type of sensilla is divided into two sub-types based on their morphology, especially the base circular socket and blunt or sharp tip.

Sch1 were widely distributed on the dorsal, lateral, and ventral sides of each segment, except at the termini of the antennae (Figure 2a,c,e,j and Figure 3a,e,f). This kind of sensillum is inserted into a flexible socket and has longitudinal grooves on the surface. Moreover, these sensilla have a cuspidal and bent tip. A pore is situated on the tip of this kind of sensillum. Sch1 is 43.68 μm long in female *M. usitatus*. However, female *T. palmi* Sch 1 is shorter than 30 μm (Table 3). The number of these sensilla varies from 2 to 10 and from 6 to 8 on each antennae segment of *M. usitatus* and *T. palmi*, respectively (Table 2).

Sch2 has no basic sockets and gradually tapers into the blunt tip. This type of sensillum is mainly distributed on the last flagellum sub-segment (Figure 2i and Figure 3h). The length of these sensilla is 32.05 μm long in female *M. usitatus* (Table 3). The proximal, middle and distal diameter of these sensilla is 1.73 μm, 1.57 μm and 0.82 μm, respectivey in female *M. usitatus* (Table 4). In female *T. palmi* Sch 2, it is shorter than 20 μm (Table 3). 

#### 3.2.5. Sensilla Styloconica (Sst)

These sensilla were found on the third and fourth sub-segments of the flagellum (Table 2). They are perpendicular to the antennal surface and have a smooth basal portion. These sensilla expand parallel to the antennal axis, and the median part is covered with deep grooves, finally leading to a sharp tip (Figure 2i and Figure 3g,i). The length of the female *M. usitatus* Sb1 (11.37 μm) is much longer than that of the female *T. palmi* Sb1 (8.55 μm) (Table 3).

#### 3.2.6. Sensilla Basiconica (Sb)

Sensilla basiconica were identified only on the flagellum (Table 2). They are multi-porous sensilla with shallow, longitudinal grooves and pores throughout their walls. Most of these sensilla are not curved, but some are slightly curved. The three types of this sensillum are distinguished from each other through differences in external morphology.

Sb1 is characterized by a long, straight or slightly curved peg with a sharp tip. They are U shaped with two arms and insert into a wide depressed or sunken surface. The walls are covered with abundant pores. These sensilla are distributed on the terminal of the first and second flagellum sub-segment (Figure 2e,g and Figure 3b,e). The length of female *M. usitatus* Sb1 (36.35 μm) is much longer than that of female *T. palmi* Sb1 (16.16 μm) (Table 3).

Sb2 are short, straight, and have an inflexible socket at the base. This type of sensillum is only present on the ventral aspect of the second flagellum sub-segment, close to Sb1. These sensilla are perpendicular to the antennal surface (Figure 2e,h). The length of female *M. usitatus* Sb2 is 8.15 μm, and the length of female *T. palmi* Sb2 is 5.77 μm (Table 3).

The base of Sb3 lies tightly on the cuticle of the antenna, and its tip is slightly sharp. These sensilla were discovered on the lateral side of the third, fourth, and fifth flagellum sub-segments of *M. usitatus* but were absent on the fourth sub-segment of *T. palmi* (Figure 2i and Figure 3c,g).

#### 3.2.7. Sensilla Cavity (Scav)

Sensilla cavities (Scav) are circular cavities formed as a result of the invagination of the antennae surface. This kind of sensillum is distributed on the first sub-segment of flagellum in both species (Figure 2e,f).

## 4. Discussion

In Thysanoptera, antennal sensilla have diverse shapes and are highly structurally modified, even in different species within the same genera [4]. A division in subtypes was difficult given their variability in shape at the same position [5]. Recently, Ding [22] identified four subtypes of sensilla basiconica in the adults of *F. occidentalis* based solely on their shape, but De Facci [5] thought three of these subtypes (BIII, a putative DW, BIV and BV) may be sensilla chaetica. In addition, Ding [22] stated the presence of U-shaped sensilla, but Zhu [4] and Li [23] both named this morphological type of sensillum sensilla basiconica. A recent study on the antennal sensilla of *Scirtothrips dorsalis* Hood found, using TEM, that sensilla chaetica II had a relatively thick wall and lumen and was innervated by a single unbranched dendrite that terminated in a tubular body. This suggests that this type of sensilla has a role as a mechanoreceptor [4]. De Facci [5] considered that the presence of sensilla chaetica in high proportions, even dorsally, is advantageous for thrips in probing the landscape for the perfect place to hide. A recent study has shown that Böhm bristles control antennal positioning in the *Daphnis nerii* moth via a reflex mechanism [4,24]. Furthermore, most of the studies on the sensilla of Thysanoptera provide detailed descriptions of only the external morphology; the ultrastructural evidence required for the prediction of the sensory function is relatively scarce [4,5,22,23]. This is probably because of difficulties in the fixation process for TEM and the tiny size of thrips. Investigations of their fine structure are needed. The sensilla basiconica could represent the starting point for electrophysiological investigations using plant VOCs as stimuli [5].

## 5. Conclusions

Here, we provide the first complete external morphological investigation of antennal sensilla in *M. usitatus* and *T. palmi*. The antennae of both species are moniliform in shape and consist of a scape, pedicel, and flagellum. It is noted that the flagellum of *M. usitatus* is made up of six sub-segments, whereas the flagellum of *T. palmi* consists of five sub-segments. We categorize the antennal sensilla of the *M. usitatus* and *T. palmi* into seven different types, based on morphology, as investigated via SEM: Böhm bristles (BB), sensilla campaniformia (Sca), sensilla trichoid (St), three types of sensilla basiconic (Sb1, Sb2 and Sb3), two types of sensilla chaetica (Sch1 and Sch2), sensilla styloconica (Sst) and sensilla cavity (Scav). The scape and pedicel exhibit Sch1, BB and Sca. The flagellum exhibit two types of Sch, three types of Sb, St, Sst and Scav. The function of each type of sensillum in *M. usitatus* and *T. palmi* requires further study, with a particular focus on the electrophysiological characteristics of the sensillum.

## Figures and Tables

**Figure 1 insects-10-00251-f001:**
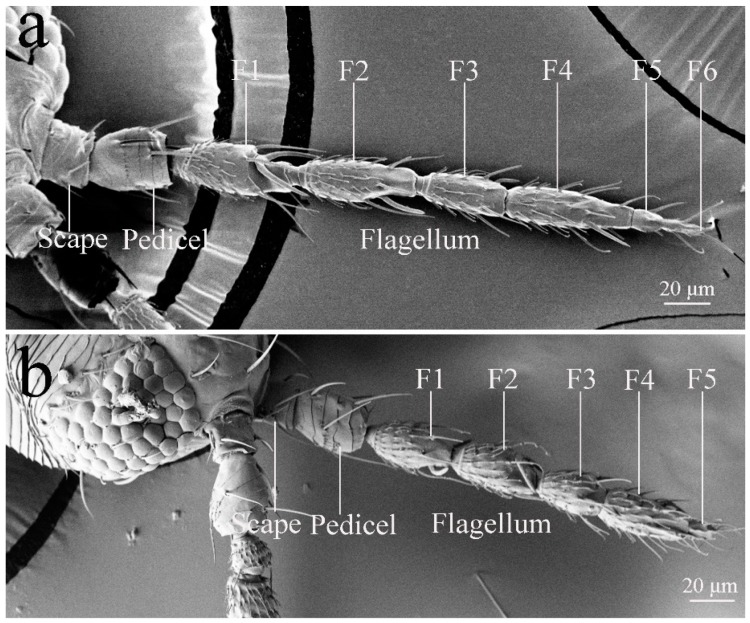
Morphology of antennae in female Megalurothrips usitatus and female Thrips palmi. (**a**) Megalurothrips usitatus. (**b**) Thrips palmi.

**Figure 2 insects-10-00251-f002:**
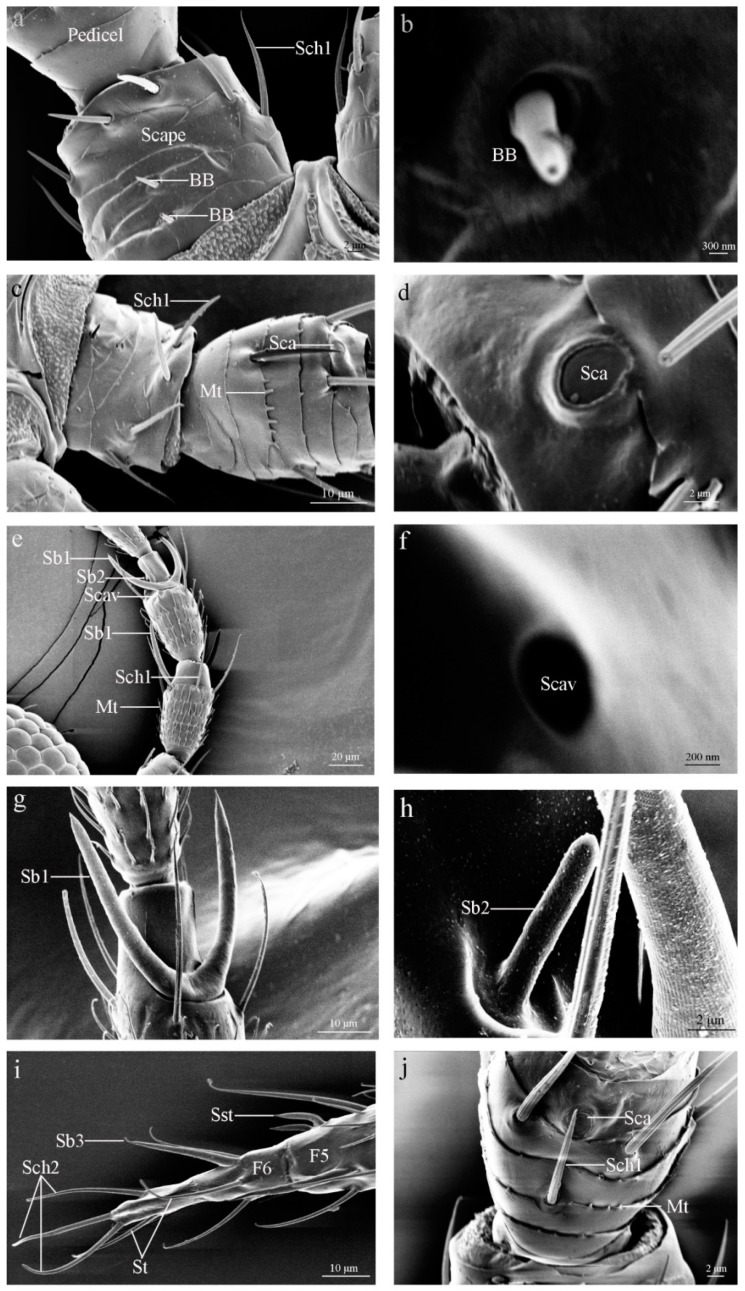
Scanning electron microscopy micrographs describing the different features of the antennae sensilla of *Megalurothrips usitatus* (Bagnall). (**a**) Böhm bristle (BB) and sensilla chaetica 1 (Sch1) were found on scape. (**b**) The morphology of Böhm bristle (BB). (**c**) Sensilla campaniformia (Sca), sensilla chaetica 1 (Sch1) and microtrichia (Mt) were found on scape and pedicel. (**d**) The morphology of sensilla campaniformia (Sca). (**e**) Sensilla cavity (Scav), sensilla chaetica 1 (Sch1), sensilla basiconica 1 (Sb1), sensilla basiconica 2 (Sb2) and microtrichia (Mt) were found on the F1 and F2 of flagellum. (**f**) The morphology of sensilla cavity (Scav). (**g**) The morphology of sensilla basiconica 1 (Sb1). (**h**) The morphology of sensilla basiconica 2 (Sb2). (**i**) Sensilla chaetica 2 (Sch2), sensilla basiconica 3 (Sb3), sensilla styloconica (Sst) and sensilla trichodea (St) were found on the F5 and F6 of flagellum. (**j**) Sensilla campaniformia (Sca) were found on pedicel.

**Figure 3 insects-10-00251-f003:**
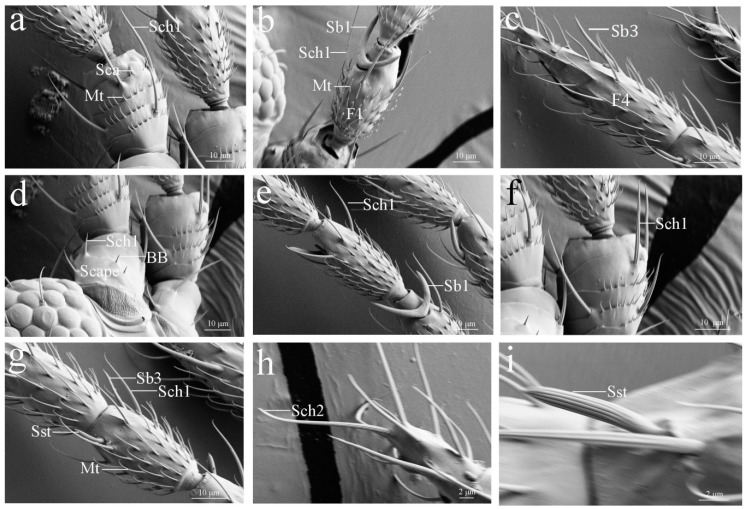
Scanning electron microscopy micrographs describing different features of the antennae sensilla of *Thrips palmi* (Karny). (**a**) Sensilla campaniformia (Sca), sensilla chaetica 1 (Sch1) and microtrichia (Mt) were found on the scape and pedicel. (**b**) Sensilla basiconica 1 (Sb1), sensilla chaetica 1 (Sch1) and microtrichia (Mt) were found on the F1 of flagellum. (**c**) Sensilla basiconica 3 (Sb3) was found on the F4 of flagellum. (**d**) Böhm bristle (BB) and sensilla chaetica 1 (Sch1) were found on the scape. (**e**) Sensilla basiconica 1 (Sb1) and sensilla chaetica 1 (Sch1) were found on the F1 and F2 of flagellum, respectively. (**f**) The morphology of sensilla chaetica 1 (Sch1). (**g**) Sensilla styloconica (Sst), sensilla basiconica 3 (Sb3) and sensilla chaetica 1 (Sch1) were found on the F3 of flagellum. (**h**) The morphology of sensilla chaetica 2 (Sch2). (**i**) The morphology of sensilla styloconica (Sst).

**Table 1 insects-10-00251-t001:** Mean length (μm) of antennal segments in *Megalurothrips usitatus* and *Thrips palmi*.

Specie	Gender	Scape (μm)	Pedicel (μm)	Flagellum (μm)						Total Length (μm)
				F1	F2	F3	F4	F5	F6	
**M**	female (*n* = 10)	32.79 ± 0.63	37.43 ± 0.68	55.38 ± 0.97	59.43 ± 1.34	38.07 ± 1.08	58.02 ± 0.33	15.25 ± 0.49	23.47 ± 0.44	319.84 ± 2.17
	male (*n* = 10)	21.62 ± 0.96	30.53 ± 0.66	44.30 ± 0.93	47.77 ± 1.98	30.90 ± 1.03	48.09 ± 1.43	12.06 ± 0.46	16.22 ± 1.86	252.66 ± 6.71
T	female (*n* = 10)	17.35 ± 1.31	30.50 ± 0.99	38.50 ± 0.86	39.01 ± 1.26	31.19 ± 0.52	43.04 ± 0.80	14.31 ± 0.34	-	213.89 ± 1.92
	male (*n* = 10)	17.63 ± 0.73	31.58 ± 1.77	36.06 ± 2.40	39.60 ± 1.02	30.88 ± 1.21	42.38 ± 1.68	14.76 ± 0.69	-	212.88 ± 3.22

The columns were mean ± standard error. M and T in the first column stand for *M. usitatus* and *T. palmi*, respectively. “-”stands for no situation.

**Table 2 insects-10-00251-t002:** Number and distribution of antennal sensilla of *Megalurothrips usitatus* and *Thrips palmi.*

Antennal Segments		Species (*n* = 10)	BB	Sca	St	Sst	Scav	Sch		Sb		
								1	2	1	2	3
**Scape**		**M**	5	-	-	-	-	10	-	-	-	-
		T	5	-	-	-	-	8	-	-	-	-
**Pedicel**		M	-	1	-	-	-	7	-	-	-	-
		T	-	1	-	-	-	8	-	-	-	-
**Flagellum**	F1	M	-	-	-	-	1	6	-	1	-	-
		T	-	-	-	-	1	6	-	1	-	-
	F2	M	-	-	-	-	-	6	-	1	1	-
		T	-	-	-	-	-	6	-	1	1	-
	F3	M	-	-	-	1	-	6	-	-	-	1
		T	-	-	-	1	-	6	-	-	-	1
	F4	M	-	-	-	1	-	8	-	-	-	2
		T	-	-	-	1	-	6	-	-	-	-
	F5	M	-	-	-	-	-	2	-	-	-	1
		T	-	-	4	-	-	-	4	-	-	1
	F6	M	-	-	2	-	-	-	3	-	-	-

“-” stands for no situation.

**Table 3 insects-10-00251-t003:** Mean length (μm) of antennal sensilla of female *Megalurothrips usitatus* and *Thrips palmi.*

Species	BB	St	Sst	Sca	Sb			Sch	
					1	2	3	1	2
M (*n* = 10)	3.04 ± 0.12	19.34 ± 1.53	11.37 ± 1.10	3.86 ± 0.13	36.35 ± 2.64	8.15 ± 1.23	27.55 ± 2.46	43.68 ± 1.56	32.05 ± 2.16
T (*n* = 10)	2.59 ± 0.16	-	8.55 ± 0.22	3.75 ± 0.17	16.16 ± 0.73	5.77 ± 0.35	-	26.56 ± 1.11	16.31 ± 0.61

The columns were mean ± standard error. M and T in the first column stand for *M. usitatus* and *T. palmi*, respectively. “-“ means this value was not measured.

**Table 4 insects-10-00251-t004:** Mean diameter (μm) of antennal sensilla of female *Megalurothrips usitatus* and *Thrips palmi.*

	Species (*n* = 10)	BB	St	Sst	Sb			Sch	
					1	2	3	1	2
Proximal	M	0.81 ± 0.04	1.16 ± 0.09	3.98 ± 2.05	4.31 ± 0.33	2.01 ± 0.22	3.12 ± 0.30	-	1.73 ± 0.13
	T	-	-	1.24 ± 0.04	0.89 ± 0.05	-	-	-	-
Middle	M	0.38 ± 0.13	-	1.39 ± 0.13	3.10 ± 0.20	1.75 ± 0.27	2.13 ± 0.19	-	1.57 ± 0.12
	T	-	-	1.27 ± 0.06	-	-	-	-	-
Distal	M	-	-	1.11 ± 0.11	1.26 ± 0.11	0.92 ± 0.14	1.38 ± 0.18	-	0.82 ± 0.09
	T	-	-	-	-			-	-

The columns were mean ± standard error. M and T in the first column stand for *M. usitatus* and *T. palmi*, respectively. “-“ means this value was not measured.

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
