# Peer review of "Morphology and Distribution of the Antennal Sensilla of Two Species, Megalurothrips usitatus and Thrips palmi (Thysanoptera: Thripidae)"

_insects, 2019, doi:10.3390/insects10080251_

Round 1
Reviewer 1 Report
The manuscript has improved considerably and the authors have taken care of earlier criticism. The EM photos are now well organized, but a general error is that the subfigures are referred to by their capital letter in the text, but in the figures lower case characters are used. These have to match! Errors/comments by line number:
8: W.
16: economically important (without hyphen)
24: basiconica
45: respond
92: measured
94: refer
98: delete first sentence
103-4: Since you write “approximately”, I suggest lengths are given without decimals, i.e. 320, 253, 214, 213.
105, Table 1: I suggest to add a sentence on line 105 saying that there was a consistent sexual difference (p<0.05) in M. s. for all antennal segments and total length, but not so in T. p. At the same time, all the letters a and b can be deleted from the table, but a statement as above added.
117: sensilla are
130: sensilla reached slightly
135: antennal
curved138: sensilla is
142: Delete first sentence (not relevant)
178: “abundant” is not a correct description. Maybe you mean “thick” but then you have to make TEMs. I suggest you skip this statement.
195, 201: De Facci
206: Also ref. #5 contains a number of good TEM micrographs
Figure 3: “f” might be seen better if black?
247: De Facci
261: Hymenoptera
272-4: I cannot find this paper in J Zool, but in Journal of Biology, Agriculture and Healthcare. 2013, 3(8), 81-86.
276: Trop.
283: Schneider, D.
284: Zacharuk, R.Y.
289: ; instead of , between authors
Author Response
Dear Editors and Reviewers:
Thank you very much for your letter and the comments concerning our manuscript entitled “Morphology and Distribution of the Antennal Sensilla of Two Species, Megalurothrips usitatus and Thrips palmi (Thysanoptera: Thripidae)” (Manuscript ID: insects-574832). Those comments are all valuable and very helpful for revising and improving our paper, as well as the important guiding significance to our researches. We have studied comments carefully and have made correction which we hope meet with approval. Revised portion are marked in red in the paper. Please see the attachment.
Special thanks to you for your good comments.
We tried our best to improve the manuscript and made some changes in the manuscript. These changes will not influence the content and framework of the paper. And here we did not list the changes but marked in the revised paper.
We appreciate for Editors/Reviewers’ warm work earnestly, and hope that the correction will meet with approval.
Once again, thank you very much for your comments and suggestions.

Reviewer 2 Report
Please see the attachment.

Author Response
Dear Editors and Reviewers:
Thank you very much for your letter and the comments concerning our manuscript entitled “Morphology and Distribution of the Antennal Sensilla of Two Species, Megalurothrips usitatus and Thrips palmi (Thysanoptera: Thripidae)” (Manuscript ID: insects-574832). Those comments are all valuable and very helpful for revising and improving our paper, as well as the important guiding significance to our researches. We have studied comments carefully and have made correction which we hope meet with approval. Revised portion are marked in red in the paper. Please see the attachment.
Special thanks to you for your good comments.
We tried our best to improve the manuscript and made some changes in the manuscript. These changes will not influence the content and framework of the paper. And here we did not list the changes but marked in the revised paper.
We appreciate for Editors/Reviewers’ warm work earnestly, and hope that the correction will meet with approval.
Once again, thank you very much for your comments and suggestions.

This manuscript is a resubmission of an earlier submission. The following is a list of the peer review reports and author responses from that submission.
Round 1
Reviewer 1 Report
Review of “ A comparative study on structure and main 2 characteristics of antennal sensilla in two thrips 3 species, Megalurothrips usitatus and Thrips palmi 4 (Thysanoptera: Thripidae)” by Wang et al.
The manuscript is a rather straightforward description of the antennal setup of different sensilla in two species of thrips, which are known pests on different crops. In general the study seems to be well performed and the presentation makes sense. However, there are a large number of issues that have to be corrected before the manuscript is possible to publish, both in terms of content and language.
Content
Line 67: I don’t think it is correct to say that the study will provide information about the host acceptance mechanism, as nothing of this behaviour is studied. Please modify the writing.
Line 71-2: Please provide scientific names of the host plants.
Line 80: What city for SZ760?
Line 96: How were various lengths (segments, sensilla) measured, any tool or program?
Results and Tables: Are you really sure you have measured each sensillum to the exact nm? If not, you have to reduce the number of decimals, so they all make sense.
Tables: Although mentioned in the text, you should give the n-values in each table. Also, it is not clear if the measurements of sensilla were on male or female or both.
Line 113-4: What is meant by “significant variation”. Were they different or not?
Line 145: It would be good to show some “tips” in a picture.
Line 252-3: Nine types, but many are not chemical detectors, so nine types do not necessarily mean a complex olfactory system.
Line 260: No electrophysiological studies have been made, so you cannot talk about “further” studies. I guess you mean “future”.
Figures: Fig. 1 is a bit dark and not as sharp as Fig. 2 (at least on my screen). It would be good if this could be improved. Both figures have an unacceptable sloppy layout. Please make all spaces between pictures of the same width. Put sub-figure letters inside each picture. Also, the direction of all arrows is in opposite direction compared to the normal use. I think this is disturbing to most readers and should be changed.
Legends: Should be “trichodea” and “basiconica”
Language
Although it is possible to understand most of the text, it is in bad need of a linguistic check by a native English-speaking person. I can only point out some examples here:
Line 16: Insert “the” before “family” and delete “the” after “are”. Thus, the use of “the” is in many places not correct.
Line 19: Change “were” tor “are”, as it describes what is presented in the existing study.
Line 21 and 32: Following the same argument as above, change “was” to “is”. The choice of tempus, past or present, is an issue to take care of throughout the manuscript.
Line 27: Revise to “in both species. In addition, the abundance… of the above…”
Line 29: Should be “Based on these results…”
Line 41-2: Should be “The melon….to the families
Line 63: Should be “how thrips locate…”
Line 91 and many other places: Should be “antennal sensilla” not antenna or antennae sensilla or segments
Line 124-7: The names of different sensilla should not be with capital first letter, except Böhm bristle (as Böhm is a person’s name). It should be sensilla trichodea. I don’t think you can write sensilla cavity or sensilla spore, as these are not in latin or an established way of writing. My guess is that you can use cavity sensilla and spore sensilla, but please check reference literature. Microtrichoid is probably ok, or sensilla microtrichodea.
Line 153-4: It is not clear what is different and what is similar. Please reformulate!
Line 166: “except” is better than “but”
Line 168: remove “grooves” or “ridges”. Mentioning both is redundant.
Line 204: Name in italics, please.
Line 219-20: “detected on the dorsal,…ventral sides… “
Line 223-4: “…antennae of both…consisted of three basic segments, …”
Line 238: “s. trichodea” not in italics!
Line 247: “… were similar to…”
References: This is a mess! Please follow instructions of the journal and be consistent! Some examples: No capital letters of words inside titles like in refs 1 and 2. Journals abbreviated or not. Delete issues if they are redundant. “;” or “,” before pages? Ref. 22, please provide pubisher.
Reviewer 2 Report
1. Although generally understandable, the text would benefit from
correction by someone fluent in English. For example, on the first page,
the sentence at lines 41-42 has no verb. The reference 4 at line 43 is
clearly incorrect.
2. Lines 49-55 are essentially “motherhoods” – where is the experimental
evidence that the antennae of these thrips detect any chemical signals?
The references 13-16 do not support the statements given – 13 deals with
positional sensoria in the pedicel; 14 is irrelevant; 15 deals with
sensilla in the gut; 16 is irrelevant.
3. Lines 55-57, the authors rely heavily on publications from China. 17
appears to be a thesis, and 18 I have found difficult to access. Why are
there no reference to publications on thrips antennal sensoria in major
journals, such as:
Slifer, E.H. & Sekhon, S.S. (1974) Sense organs on the antennae of two
species of thrips (Thysanoptera, Insecta). Journal of Morphology, 143,
445–456.
De Facci, M.D., Wallén, R., Hallberg, E. & Anderbrant, O. (2011)
Flagellar sensilla of the eusocial gall-inducing thrips Kladothrips
intermedius and its kleptoparasite, Koptothrips dyskritus (Thysanoptera:
Phlaeothripinae). Arthropod Structure & Development, 40, 495–508.
4. Lines 60-61. Where is the data concerning function to support this
assertion?
5. Lines 62-68 These again seem to be the typical sweeping statements of
a student trying to impress a supervisor, but with little precise thought.
6. Lines 107-111 concerning antennal movements. These statements are
contrary to the experience of this reviewer in watching the behaviour of
thrips. Antennal segments are not “passive in relation to the pedicel” –
by which the authors presumably imply that the segments cannot move
independently. In life thrips curve their antennae when exploring their
food plant or when mating.
7. Lines 113-118 Antennae of females are stated to be longer than those
of males, but this is due to the fact that in both these species females
are much larger than males. There is thus sexual dimorphism in body
size, but contrary to lines 227-229 there is no evidence of sexual
dimorphism in the antennae
8. Each sensillum is supplied with a nerve, and for most sensilla this
is a single nerve. The authors provide no evidence that some of the
“sensilla” they list have any nerve supply. In particular, microtrichia
are rigid, non-articulated prolongations of the cuticular surface and
have no nerve supply – they are NOT sensilla. Similarly, the structures
the authors term “Sensilla spore” [I do not understand the derivation of
that curious term] – are considered by other thrips workers to be
microtrichia. I also have doubts about the poorly defined structures
termed “sensilla cavity”, and suspect these may be artefacts.
9. Lines 223-226. “This study demonstrated….” – To be more precise, this
study confirmed something that has been known for well over 100 years.
229-230 – there are good references to antennal sexual dimorphisms in
various genera of thrips. NB the reference numbers in these paragraphs
are incorrect.
10. Lines 233-237. The authors do not seem to realise that the two setae
on the dorsal surface of the first antennal segment (the scape) that
they refer to as “Bohm bristles” are mirrored by a second pair on the
ventral surface. Moreover, these two pairs of setae are part of the
basic body plan of Thysanoptera, and can be found in species of both
sub-orders, Terebrantia and Tubulifera.
11. Lines 237-239 These sentences do not make sense.
12. The technical work is of a high standard, and the SEM images are
excellent, but the assumptions concerning function have no obvious
support, and I am not convinced by the classification into different
sensilla. I consider that the attempts to make this study relevant to
our understanding of thrips biology are superficial and unconvincing.
Moreover, there is no serious attempt to place the study within the
context of comparative morphology of thrips and related insect groups.
Reviewer 3 Report
Findings of the study are significant and worth publishing, and authors have done great work in the presentation of the research. My only concern is with the language used- the text of the paper needs to be thoroughly revised by a native English speaker before it can be accepted for publication.